# Long-Term Care Insurance for Older Adults in Terms of Community Care in South Korea: Using the Framework Method

**DOI:** 10.3390/healthcare12131238

**Published:** 2024-06-21

**Authors:** Yongho Chon, Seok-Hwan Lee, Yun-Young Kim

**Affiliations:** 1Department of Social Welfare, Incheon National University, Incheon 22012, Republic of Korea; chamgil@inu.ac.kr; 2Seoul 50 Plus Foundation, Seoul 04147, Republic of Korea; seokhwan@50plus.or.kr; 3Department of Social Welfare, Jeonbuk National University, Jeonju 54896, Republic of Korea

**Keywords:** community care, long-term care insurance, care service, service delivery system

## Abstract

This study aimed to analyze the long-term care insurance for older adults in South Korea in terms of community care. An analytical framework was designed for the study, focusing on comprehensiveness, adequacy, and integration. The findings suggest that Korean LTCI is significantly limited for the development of community care. First, in terms of comprehensiveness, the use of visiting nurses and the availability of short-stay services have been significantly reduced, and the supporting services for informal caregivers are at the beginning stage of their introduction. Second, in terms of adequacy, the quantity of benefits, such as three to four hours of care a day, are insufficient to meet older adults’ substantial needs. Furthermore, the overall quality of home care services is low, particularly with regard to short-stay services and welfare equipment. Finally, in terms of integration, basic linkage of organizations has not been properly conducted in local areas, and there remains an absence of care managers in the LTCI system. To cope with these challenging issues, the following policy measures are suggested: the activation of rehabilitation services, the expansion of benefit quantities, the improvement of service quality, and the creation of organizational linkages through local authorities and long-term care centers.

## 1. Introduction

To achieve the principle of aging-in-place for the elderly, community care has been emphasized as an important policy paradigm for care policies in many countries, including advanced nations. The core philosophy of community care is that older adults should live in their homes and communities and avoid institutionalization for as long as possible, thereby aiming to reduce the health budget usage [1,2]. In advanced nations, the rate of hospital bed usage has significantly decreased due to community care policies such as home- and community-based services (e.g., −22.4% in Denmark and −21.7% in Finland between 2009 and 2019).

Following the community care policy of advanced countries, South Korea introduced a new long-term care insurance (hereafter LTCI) system with a home- and community-based care system for the elderly in 2008. This system is a social insurance separate from the National Health Insurance. Despite this, South Korea has still shown a rapid increase of 25.6% in hospital bed usage, which is the second highest among the OECD countries after Luxembourg [3]. This indicates that the LTCI systems have not been sufficiently constructed in South Korea to cater for the characteristics of the community, leading to high usage rates of long-term care (hereafter LTC) hospitals and inefficient budget use and waste. In 2021, the expenditure on LTCI was USD 7439 trillion; however, the health insurance expenditure at LTC hospitals was as high as USD 4471 trillion [3].

Community care is an urgent service that aims to ensure humane and dignified aging and financial sustainability for older adults. To achieve this, a decent community care system should be developed. For instance, a number of diverse services should be provided in an adequate and integrated way to many older people in need of care [4,5]. The Korean government has implemented a series of pioneering projects of community care in some local areas. The Korean LTCI is a core care service that was used by 10.1% (899,113 persons) of the total older adult population (8,912,785 persons) in 2021 [6]. Therefore, we need to examine the reality of the Korean LTCI from the perspective of elderly users. In particular, elderly users suffer from a decreased ability to perform activities of daily living (ADL), which makes independent living difficult and leads to a need for LTC or admission to LTC hospitals or nursing homes. This group is a direct target of community care, which is intended to keep patients out of nursing homes. We should ask the following from the perspective of local users: ‘What can we do to fulfill people’s unmet needs at home and in the community for as long as possible, and to delay the time until they have to be admitted to a facility?’.

Recently, there has been an increase in the number of studies focusing on the concept of community care [7], macroscopic policy measures such as related overseas policies [8], and case studies of pilot projects [2,9,10]. However, few studies directly connect community care and LTCI for older adults and analyze their relationship in depth. Although some pilot projects of community care have been conducted in some areas, they tend to develop new kinds of services, such as meals-on-wheels, and do not link the pilot projects with the LTCI system in an integrated way [10,11,12]. Many advanced nations operate LTC in an integrative manner as both a core system and mechanism of the community [13]; however, South Korea’s approach has been segmented.

Given this, the study aims to examine the Korean LTCI service delivery system for the elderly in terms of community care and to explore the necessary improvements that can aid in achieving the objectives of community care. To do this, we presented an analytic framework [14] in which we operationally defined the principles of a service delivery system suited to the intentions of community care and then diagnosed the state of the LTCI for older adults in South Korea. We conducted a literature review and assessment of the Korean LTCI using a qualitative framework method. This provided an overall picture of the Korean LTCI service delivery system in terms of community care, and some useful policy lessons will be highlighted.

## 2. Methods

### 2.1. Target of the Research

This study examines the Korean LTCI service delivery system. As shown in Figure 1, the elderly must access and use the service in accordance with the service delivery system. The National Health Insurance Corporation (NHIC), who is the insurer, is responsible for most of the roles in the LTCI system. Therefore, the elderly should apply to the NHIC and receive an eligible grade. Due to the absence of care management, the elderly or their caregivers themselves must choose the types of services they want and directly contact service providers. The service user is required to pay 15 percent of the total cost when using home care services.

### 2.2. Framework Method

The authors conducted qualitative research by adopting the framework method. As Goldsmith, Gale, and colleagues argued, ‘the framework analysis method’ has been popularly used for the explicit purpose of analyzing ‘qualitative data’ in applied research, including in health studies [14,15]. In particular, the framework method is an excellent means for the thematic analysis of academic issues. Two main components of framework analysis are creating an analytic framework and applying this framework [14]. When creating an analytic framework, the authors need to familiarize themselves with the existing literature and to identify and develop a framework given the aim and characteristics of the research [14,15]. The framework is composed of themes or concepts with subgroups relating to the issue.

Following this method process, the three authors reviewed the existing literature on the service delivery system and LTCI and developed a framework for this research given the unique characteristics of community care. We established concrete criteria for selecting literature with credible references, such as relevance to the research question, articles from peer-reviewed journals, and papers issued by credible public and private organizations (e.g., government, research institutes) [14,15]. The literature selection period ranged from 2015 to 2023. In our search for existing literature related to the topic, we utilized Korean journal search engines (namely, DBPIA and KISS), inputting keywords such as ‘long-term care insurance’, ‘community care’, ‘service delivery system’, ‘care service for the elderly (older adults, older people)’, and ‘home care’. We identified approximately 50 academic journals and papers from governmental and private organizations, and based on our selection criteria, we selected about 25 papers. All three authors deliberated on which papers to include in the study by using a star rating system. When we were not able to reach a consensus, we further discussed the differing perspectives of each author and the selection criteria in depth.

Reviewing the existing literature, the key principles of the service delivery system are summarized as follows: comprehensiveness, accessibility, sustainability, adequacy, integration, equity, professionalism, responsibility, instantaneity, and so on [16,17]. To identify and develop our research framework, the authors frequently discussed the key principles to adopt based on our research aim and policy context as the framework method suggested [14].

Through our extensive review of the existing literature, we developed our research framework as shown in Table 1. Given the unique characteristics of community care and the boundary of the research, we decide that the Korean LTCI service delivery system should be analyzed in terms of its comprehensiveness, adequacy, and integration, which are important principles of delivery systems [17,18,19]. The rationale behind this is that community care should not be vaguely pursued simply because it is politically desirable, as several problems can arise if older adults are not provided with suitable services personalized to their individual needs. If the diverse needs of older adults are not comprehensively fulfilled at home, the service provision is insufficient, or the services are not effective, immediate difficulties can arise due to the care deficit or care poverty [20]. Whereas basic needs and various services are provided in one place at LTC facilities, insufficient services at home can lead to difficulties in the lives of older adults. As such, multifaceted conditions must be met to implement community care.

Three main components of the framework are defined operationally as follows. First, we define ‘comprehensiveness’ as the provision of ‘various types of health and social care services’ to meet the needs of older adults requiring LTC at home [13]. This definition is based on the literature, which suggests that several services must be provided in the LTC system to meet the diverse needs of users at home [19]. The scope of services in this study primarily includes health and social care services at home and support services for family caregivers, which are popular in Europe [13,21]. If various services are not available to respond to the frequent needs of older adults, a structural care deficit can arise, leading to deterioration, rather than maintenance or improvement, of the functional state of individuals requiring LTC.

Second, we operationally define ‘adequacy’ as the appropriate provision of LTC services in terms of ‘quantity’ and ‘quality’ in meeting the needs of older adults [22]. For our study, we restrict the definition of adequate coverage to refer to ‘time’, and the ‘quality’ of services is evaluated with regard to the provider. Although adequacy means meeting the needs of the users through the suitable provision of services, there is scope for multifaceted discussion regarding the frequency, duration, accessibility, efficiency, and unit cost of services [22].

Third, ‘integration’ is operationally defined as the provision of multiple services for older adults via ‘linkage’, at the regional level, between different organizations related to either older adult care services or systems [23]. Integration means that users with various needs can access their required services conveniently, without visiting multiple organizations, or while using the minimum amount of organizations and staff [18,23]. Integration can be categorized into linkage at the lowest level, coordination at the middle level, and integration at the highest level, depending on the extent and methods [23,24]. However, given the low integration of delivery systems in South Korea [24], we focus on whether there is a ‘linkage’ between organizations related to LTCI for older adults.

By adopting this framework, we extensively reviewed the existing literature. When analyzing the data, three authors assessed the categories according to the definitions. If there were incongruent opinions on the ratings, the three authors discussed the issues together in depth and endeavored to reach an agreement.

## 3. Results

We reviewed approximately 25 papers relating to the research topic and assessed the issues using the framework as follows:

### 3.1. Results of the Comprehensiveness Analysis

First, there is overall a severe shortage of health-related services [25]. In particular, rehabilitation-related services are not covered. As a result, the rehabilitation needs of older adult patients suffering from physical contracture and paralysis due to stroke and the professional rehabilitation needs of older adult patients with cognitive dysfunction, such as dementia, are not being met. When assessing the needs for the selection of service recipients, rehabilitation needs are a major assessment category; however, the kinds of services required to meet such needs are lacking. Given that advanced nations, including the United States, Japan, and Europe, are actively using rehabilitation services to maintain or improve older adults’ health and function [13], South Korea is significantly behind in this area. The lack of rehabilitation service provision is incurring social costs of approximately USD 864 million annually [24]. In particular, no measures have been taken to remedy the situation for older adult patients who are de-institutionalized from LTC hospitals or nursing homes and who experience a decline in physical and mental function, such as loss of muscle strength and stamina due to a restricted range of activities and long periods in bed.

Second, the use of visiting nursing services is severely lacking. Currently, the only healthcare service provided in LTCI for older adults is visiting nursing, but this accounts for only 0.5% of the home care services (based on NHIC payments in 2021) [6]. In Western countries, visiting nursing services are one of the main services used to systematically manage multiple chronic diseases in LTC patients [13]. However, in South Korea, the use of visiting nursing services has continually declined since the early implementation of the policy. This is because there is no systematic device to mandate the usage of visiting nursing services, even when the patient has nursing needs, such as bedsores. Users and caregivers can freely select their required services, irrespective of their actual needs, behind the excuse of giving choices to the users. The neglect of visiting nursing services persists because caregivers prefer visiting (social) care services, which perform various household tasks for a relatively low price. This flawed provision of choice leads to unmet visiting nursing needs for older adults and is the cause of LTCI for older adults being run as a social care service with an excessive focus on visiting care (68.1% of all covered home care service, based on NHIC payments in 2021) [6].

Third, there has been a severe decline in short-stay services. In 2017, short-stay ser-vices accounted for only 0.5% of all home care coverage; however, this continued to decrease even further to 0.1% in 2021 (USD 5256 million) [6]. Short-stay services are essential for the vitalization of community care, as they represent a middle stage between nursing homes and home services. These services allow older adults to access the required services for a short period without being admitted to a facility, providing informal caregivers with a brief period for rest or other work before the elderly adult returns home. However, as of 2021, there were only 163 short-stay providers in South Korea and only 567 users thereof [6]. Furthermore, only 44 short-stay providers are available in Seoul, meaning that there are fewer than two facilities per district. The government has been passive in expanding short-stay providers. In practice, short-stay providers have to be managed like live-in facilities, but the fees are equivalent to home care services, and most have been converted into nursing homes. Moreover, users and caregivers have little awareness of the short-stay services, leading to problems in securing regular business [26]. Although the fees for short-stay services have increased, coverage is still low compared with nursing homes (e.g., USD 46.50 for 1 day at a Grade 1 short-stay service facility; USD 49.78 for an older adult communal living care home), and usage is limited to a maximum of nine days per month. Therefore, it is structurally difficult to provide the same level of services as other facilities.

Finally, support policies for informal caregivers in LTC, such as family counseling support services for caregivers suffering from stress and depression, and rest systems for families of older adult dementia patients (e.g., short-stay service and all-day visiting care service) are being implemented [27,28]. However, these pilot projects are only being implemented in some regions and are not service types that are officially covered under the LTCI system. Therefore, the overall usage is still severely limited. Family counseling support services are mostly used by spouses. These services were used by 2581 persons in 2019, but this decreased to 2406 persons in 2021 [6]. Although there are 179 branches of the NHIC nationwide, as of 2022, only 65 NHIC centers were providing family counseling support services. In addition, because the holiday system for families of dementia patients can only be used for up to nine days per year, it was only used by 845 persons in the first half of 2021 [28]. With short-stay service facilities gradually becoming scarcer, there is a major shortage in the provision of rest for families of dementia patients. Support services for informal caregivers are still in the early stages [28]; if community care is implemented and care roles are expanded, relatives and neighbors will also have to be included. Furthermore, among those with geriatric diseases, a holiday system is only provided for those suffering from dementia [28]; however, depending on the need for care, this will have to be expanded to a more universal system that includes other severe diseases, such as stroke.

### 3.2. Results of the Adequacy Analysis

The services provided to older adults should meet the users’ needs regarding quantity and be sufficient to allow the users to live at home and maintain a certain level of quality [29]. In terms of the adequacy of service time, we first found that visiting care was limited to a maximum of 3–4 h per day. Some advanced welfare countries do not limit the scope of coverage in order to sufficiently meet users’ needs and maintain equality for severe patients; rather, they provide a flexible, variable level of coverage [13]. However, since the implementation of the system, South Korea, like Germany and Japan, has set maximum limits to visiting care to control costs [29]. When these limits are exceeded, the user has to pay the full cost out of their own pocket. However, 3–4 h per day is insufficient to meet the needs of Grade 1–2 bedridden patients and older adult patients with severe conditions and limited mobility.

Second, we found that the maximum benefit of home care services is low, at only 55.91–91.35% of nursing home services. This is much lower than in other countries that have implemented a similar LTCI system, such as Germany (89.5–102.9%) and Japan (75.7–119.3%) [3]. In particular, Grade 1–2 patients qualify for the use of both nursing homes and home care services, but the extent of the home care benefits is lacking com pared with the nursing home coverage, causing an obstacle to needs-based, equitable service use [21]. These insufficiencies of home care benefits were found to be an important factor leading to negative outcomes, such as the early admission of older adults to nursing homes and worsening finances of the LTCI system. Moreover, the method for the provision of LTC services is uniform and inflexible for users. The LTCI services are provided based on using a certain allotment of time during a single visit. Inefficient service provision can be a problem, such as caregivers taking a rest during work or spending time on unnecessary conversation.

Third, the overall quality of home care service providers is not satisfactory. Table 2 presents the results of the quality assessments for home care providers in 2020. Visiting care providers are one of the major home care services, and 65.7% of these providers received an A (excellent) or B (good) grade, while the other 34.3% received lower grades. This represents an improvement compared with the situation in 2017 [30]. However, the service type with the lowest grades was short-stay providers, with only 7.0% receiving an A grade, and almost half (46.5%) receiving poor grades (C, D, or E). Likewise, only 42.5% of welfare device providers received an A or B grade.

When examining the evaluation results for home care providers by domain in detail, the organizational management and process of service provision domains showed relatively low scores, whereas the outcomes of coverage provision and environment and safety domains showed relatively high scores [30].

Moreover, there are major limitations relating to the care workforce, which are directly associated with the quality of services. Being a care worker is an unstable position, due to low wages, long labor times, and poor treatment [31]. The resulting high turnover rate and staffing shortages have a severe effect in practice [5]. As well as structural instability among service providers (such as excessive competition in some LTC markets, including urban regions), and falling behind due to the failure to secure clients, there is a lack of training to foster capable staff and a lack of continuing education. This lack of expert service personnel results in service provision that focuses on housework and basic care and fails to improve older adults’ actual functioning [32].

### 3.3. Results of Integration Analysis

Integration is the feature that most clearly shows the unique nature of community care; it is the extent to which several organizations in the community enable users to conveniently access several services at once through organizational linkages [18].

First, we found that LTCI for older adults in South Korea is constructed as a centralized service delivery system, and each branch of the NHIC does not attempt any basic linkages with other public health and social welfare-related organizations in the region. The NHIC’s LTC management centers, which perform the various tasks involved in LTCI, recognize LTC organizations as entities to be regulated and do not act as a hub that can link necessary resources from the perspective of older adults [24]. In addition, the majority of LTC service providers show a strong tendency toward making profits from commercialization; therefore, they are hesitant to form cooperative relationships, such as linking with other older adult care organizations based on a philosophy of regional welfare. The branches of the NHIC sometimes work in cooperation with local government at the level of county or district offices [24]. However, the purpose of this cooperation is mostly limited to managing finances by uncovering illegal claims for LTC service benefits. This is closely related to our current reality, where there is no basic structural linkage at the regional level between organizations in the medical, public health, and welfare sectors [24].

Second, linkage has not been successful due to structural limiting factors. The public welfare system (in Korean: Haengbok e-Eum) used by local governments and the data network used by LTCI for older adults are not provisionally connected, and data regarding LTC recipients cannot be easily accessed by staff at the local government department for older adult welfare [9]. Therefore, the local government has enormous difficulty in obtaining data related to LTC, and while LTCI is, in practice, operated as a separate insurance-type system, the local government is perceived as having no role or authority in this system. The NHIC has always shown a conservative attitude toward data sharing, and, as an insurer, they take the position of a type of ‘supervisor’ of LTCI. Due to this context, several current pioneering integrative community care projects have focused on preventive projects for non-graded older adults [9,10,11]. However, the users of LTCI are older adults at a stage immediately before institutionalization, and given that one of the main objectives of community care is to delay institutionalization or to support the de-institutionalization of patients, many of the patients who are borderline for institutionalization are being excluded from pioneering projects.

Third, the local government also shows a poor understanding of the characteristics and management methods of social services as a whole and displays no serious willingness to link with LTCI for older adults. This reflects a longstanding historical background in which local government has been used as a subordinate organization to conduct public aid projects directed by the central government. Because local government implements projects following guidelines set by the central government, they have been unable to autonomously plan and execute social service projects with an independent budget and authority. Furthermore, there is no incentive system to encourage LTC organizations to form a cooperative network in the field.

Fourth, care managers play a central role in LTC in advanced countries, setting up care plans using formal and informal local resources from the user’s perspective, and encouraging integrative linkage and service provision. However, this role does not exist in South Korea. Recently, the NHIC introduced Korean-style care managers by reinforcing usage support. However, the practical roles are limited to the restricted monitoring of users and the management and control of care plans compiled by service providers. If community care is to be implemented, it will be essential to consider who can perform the role of care manager, considering the various suppliers involved, including LTCI. In current pioneering projects for community care, the primary role of identifying recipients and establishing service provision plans is performed by community centers. These centers only provide simple linkage support for LTCI recipients; therefore, basic plans are required to revise this system in an integrative manner.

Finally, at a more basic level, the government has no blueprint for how to pursue the practical linkage of the public health and social welfare sectors at the regional level. To achieve practical linkage between different sectors at the regional level, this task cannot be vaguely entrusted to the region or service providers. Despite various stakeholders in different regions and sectors demanding revisions to the delivery system that favor themselves, it is essential to present systematic plans for the integrative provision of local services based on a user-centric perspective. In particular, LTCI for older adults requires institutional connections to be made by the local government with related systems, such as national health insurance and personalized care services for older adults, and with related organizations, such as public health centers, dementia care centers, and mental health welfare centers.

## 4. Discussion

As noted so far, the overall findings of the research suggest that the Korean LTCI service delivery system has some fundamental limitations in developing community care. At this point, we need to discuss the origins of such limitations within the Korean context. Above all, it seems that the Korean government has endeavored to increase the number of LTCI service users in a ‘quantitative’ way. The proportion of older adults who used the LTCI services increased from 3.1% in 2008 to 10.1% in 2021 [6]. The government has tended to proudly announce that the coverage of the LTCI system has notably increased since 2008.

However, the government has not developed the ‘software’ of LTCI in a ‘qualitative’ way. As a result, new types of services have not been introduced, and the maximum service time of 3–4 h has not significantly improved since 2008. This passive approach has had a negative impact on the development of the LTCI system in terms of comprehensiveness and adequacy. In particular, the quality of services is the ultimate outcome of services. However, the Korean government has not made significant efforts to develop it. For example, the government has not actively implemented policy measures to support the care workforce. Therefore, the shortage of home care workers has been worsening, and the salary and treatment for them are very poor [30,31], which is negatively associated with the quality of services.

Since the implementation of pioneer projects of community care in some local areas, the significance of integration between the health and social care sectors has been newly emphasized. However, as noted, the Korean government has endeavored to increase the number of LTCI service users, while the NHIC is in charge of the LTCI system, separate from local authorities. Therefore, the basic linkage between the NHIC branches, local authorities, and LTC service providers has not been formed, and the service delivery systems have been operated in a fragmented way.

Looking back on the history of the Korean LTCI service delivery system, the Korean government has had no clear concept of community care, which is quite a new concept as the implementation of pioneering projects only started from June 2019. Therefore, the key principles of the service delivery system in terms of comprehensiveness, adequacy, and integration are not well established within the Korean LTCI system.

In particular, it seems that the integration of the service delivery system is one of the most challenging principles. This is because the issues relating to comprehensiveness and adequacy can only be tackled to some extent if the central government and the NHIC have a strong will to policies that will develop the LTCI service delivery system. However, in terms of integration, linking the NHIC branches, local authorities, and service providers in local areas requires them to have regular conversations with each other and to discuss and adjust their interests under the LTCI system. However, this has rarely been attempted within the Korean LTCI system, and at the moment, this is only at the beginning stage in the majority of the local areas, except for some pioneering project areas.

## 5. Conclusions

In this study, we aimed to analyze the Korean LTCI service delivery system for older adults in terms of community care by adopting the framework research method. Our findings showed that LTCI for older adults is currently severely lacking for community care in terms of comprehensiveness, adequacy, and integration.

First, in terms of comprehensiveness, the Korean LTCI for older adults was found to have limitations in meeting diverse needs due to system insufficiency in the provision of the various services required for community care. Overall, social care focusing on visiting care was most commonly provided, and there has been a dramatic decline in previously covered visiting nursing and short-stay services [24,27]. In advanced nations, rehabilitation services for older adults with stroke or cognitive dysfunction are active [13], but these items are not covered by insurance in South Korea and so are not provided. As de-institutionalization leads to community care, home care services become even more important. Nevertheless, these services are not being introduced as part of the LTCI; and while there have been some attempts at pioneering community care projects, local governments were found to have experienced several difficulties. Moreover, since support systems for informal caregivers are still at a very early stage, the burden on caregivers will grow even further [26,28,31]. For instance, holiday systems for families of dementia patients and family counseling support services showed very limited use.

Second, in terms of adequacy, the use of visiting care services was limited to a maximum of three to four hours per day, and the maximum benefit of home care services was low, at only 55.91–91.35% of nursing home services. This is insufficient to meet the needs of discharged older adult patients with severe conditions or without informal caregivers. Furthermore, the overall quality of home care services is not satisfactory. Although the quality of visiting care services has improved slightly compared with the past, the quality of short-stay and welfare device services remains very poor [27].

Third, in terms of integration, organizations related to LTCI for older adults had not formed cooperative relations with the local government or other related local organizations. As a result, services were segmented, and even the lowest, most basic level of linkage was not achieved. In particular, the NHIC insists on working autonomously in the role of insurers and does not engage in regional linkage, although it is in charge of most of the key roles in the LTCI service delivery system. Care management systems were not being introduced due to concerns about increasing expenses. Local governments, which lacked experience in planning and promoting independent social service projects, showed a passive attitude. Within this context, there were no plans to establish LTCI for older adults within a community care framework or provide linkage with related organizations. Pioneering projects also tended to exclude older adult users of LTC with a high need for care services, focusing instead on mild older adult patients.

The findings of the research suggest a number of significant policy measures that should be undertaken to revise LTCI for older adults to meet the aims of community care. There should be an urgent need for overall reformation, including reinforcement of the LTCI system [33,34] and modifying the role of the local government. Most of all, there is a need to actively expand the new kinds of health services that can be used at home. In particular, existing services such as visiting nursing services must be revitalized, and short-stay services must be normalized. Furthermore, the expansion of region-based healthcare services, including the provision of rehabilitation services and linkage with visiting healthcare services must be pursued.

In addition, the adequacy of home care service coverage must be expanded to meet users’ practical needs, such as longer service times and more frequent home visits, and plans must be implemented to improve the service quality. In particular, service quality can be affected by key variables such as better treatment and training of the care workforce.

Finally, services are centered on local governments and the LTC centers of the NHIC. Therefore, these institutions will need to act as major hubs, linking users to relevant older adult care organizations.

In terms of the limitations of the study, we reviewed only the Korean LTCI papers concerning community care. To develop the Korean LTCI service delivery system further, it would be beneficial to study the advanced community care systems for the elderly in developed countries in the future.

## Figures and Tables

**Figure 1 healthcare-12-01238-f001:**
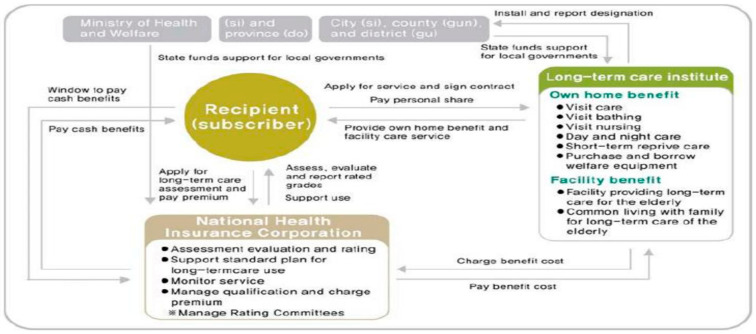
The Korean LTCI service delivery system [2].

**Table 1 healthcare-12-01238-t001:** The analytic framework of the study.

**Category**	**Definition**
Comprehensiveness	-Provision of ‘various types’ of home care services to meet the LTC needs of older adults.
Adequacy	-Provision of adequate ‘time’ and ‘quality’ of services to meet the LTC needs of older adults.
Integration	-‘Linkage’ of various home care services at the regional level to meet the LTC needs of older adults.

**Table 2 healthcare-12-01238-t002:** Quality assessment outcomes of home care providers in 2020 [30].

Category	Total		A (Excellent)			B (Good)			C (Fair)			D (Poor)			E (Fail)		
	N	Score	N	%	Score	N	%	Score	N	%	Score	N	%	Score	N	%	Score
Total	5891	84.6	2009	34.1	94.3	2014	34.2	85.7	1024	17.3	77.8	442	7.2	69.4	422	7.2	64.2
Visiting care	3540	85.3	1304	36.8	94.3	1214	34.3	85.7	594	16.8	77.7	253	7.1	69.2	175	4.9	64.4
Visiting bath	664	85.7	247	37.2	94.6	227	34.2	85.8	114	17.2	77.7	52	7.8	69.8	24	3.6	65.9
Visiting nursing	131	88.4	67	51.1	94.5	41	31.3	86.0	14	10.7	79.1	3	2.3	72.8	6	4.6	64.3
Day/night care	1053	84.2	311	29.5	94.3	401	38.1	85.7	194	18.4	77.6	82	7.8	68.6	65	6.2	65.8
Short stay	23	79.0	1	4.3	90.0	6	26.1	87.5	12	52.2	78.7	1	4.3	70.0	3	13.0	62.3
Welfare devices	480	77.9	79	16.5	94.0	125	26.0	86.0	96	20.0	78.7	31	6.5	72.8	149	31.0	63.2

## Data Availability

Data are contained within the article.

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
