# Peer review of "Long-Term Care Insurance for Older Adults in Terms of Community Care in South Korea: Using the Framework Method"

_healthcare, 2024, doi:10.3390/healthcare12131238_

Round 1
Reviewer 1 Report (Previous Reviewer 1)
Comments and Suggestions for Authors
Dear authors,
Thank you very much for significantly revising the manuscript, it is much improved. I have some general comments on the methods of this paper. It is not very clear what methods this paper used. After reading it, I think the authors conducted a literature review and assessment of the Korean LTCI based on a framework. I think it would be helpful to clarify in both the Introduction and Methods sections that the nature of this paper is qualitative research.
Another clarifying question is, given the authors rated different components of LTCI, how was the rating conducted, how many raters did the team have, and how did you reconcile incongruent opinions?
I would highly recommend the authors to revise this manuscript to reflect the aforementioned questions to make this paper more scientifically rigorous.
Author Response
Thanks to your useful comments on our paper, I was able to improve the quality of paper! We did our best to revise our paper following your comments on it. We colored blue when revising our paper.
|
|
Comments |
|
|
|
-To clarify in both the Introduction and Methods sections that the nature of this paper is qualitative research.
|
-On pager 2(line 73-74), we inserted the following sentence “We conducted a literature review and assessment of the Korean LTCI using a qualitative framework method.” - On page 3(line 91), we inserted the following sentence: “The authors conducted qualitative research by adopting framework method.” |
||
|
-Given the authors rated different components of LTCI, how was the rating conducted, how many raters did the team have, and how did you reconcile incongruent opinions? |
- On page 4(line 154-157), we inserted the following sentence: “When analyzing the data, the three authors assessed the categories according to the definitions. If there were incongruent opinions on the ratings, the three authors discussed the issues together in depth and endeavored to reach an agreement.”
|
||

Reviewer 2 Report (Previous Reviewer 2)
Comments and Suggestions for Authors
Dear Authors, Thank you very much for presenting the new version. It is much better. I think you should sort out and organise the references correctly. In the latest version, the last reference has the number 35. Also, in this version, but in some places where the unchanged text is present, there are numbers such as 37-39,42,45-46
Please see attachment for other comments.

Author Response
Thanks to your useful comments on our paper, I was able to improve the quality of paper! We did our best to revise our paper following your comments on it. We colored blue when revising our paper.
|
Comments |
Revisions |
|
- The article has 35 reference items, but the numbers are relatively high in the text, and no such papers are in the list of references. Just have a look at page 7, the second paragraph.
|
-I reduced the total number of references from 35 to 34. -The inappropriate references are changed to blue colors in the paper. |
|
- "type mistakes." Have a look at page 3, the title of part 2.2. |
-We changed the title of part 2.2 |
|
- editing the article after cleaning the references and other things I have noticed. |
- We edited the overall paper after revising the paper. |

Reviewer 3 Report (Previous Reviewer 4)
Comments and Suggestions for Authors
The changes you made improved the scientific sound of the manuscript. Thank you for your collaboration.
Author Response
Thank you for your positive comments on our paper.

Round 2
Reviewer 1 Report (Previous Reviewer 1)
Comments and Suggestions for Authors
Dear authors,
Thank you for the effort in revising this manuscript. It has been much improved and is more comprehensive. However, I think the methods section can still be improved to make it more rigorous. Please take a look at my comments below.
1. Please provide details on how the literature was searched and what databases the authors used. What were the search terms?
2. What were the selection criteria for the literature search in this study, including the year, scope, and types of studies?
3. How did the authors rate the literature? How the assessment was performed and how the authors resolved their disagreement is unclear.
4. In the Results session, first, please provide how many papers the authors included.
5. In the Discussion session, the authors need to mention the limitations of this study.
Thank you again for your efforts in revising the manuscript. Given the lack of quality research in the relevant fields, this study could provide important implications for policymakers, researchers, and practitioners. The revisions will ensure the rigor of the study.
Author Response
We Do appreciate that you gave us very useful comments on our paper. We did our best to improve our paper as follows:
- Please provide details on how the literature was searched and what databases the authors used. What were the search terms?
We changed as follows:
- In our search for existing literature related to the topic, we utilized Korean journal search engines (namely, DBPIA and KISS), inputting keywords such as ‘long-term care insurance,’ ‘community care,’ ‘service delivery system,’ ‘care service for the elderly (older adults, older people),’ and ‘home care.’(lines 107-110)
- What were the selection criteria for the literature search in this study, including the year, scope, and types of studies?
We changed as follows:
-We established concrete criteria for selecting literature with credible references, such as relevance to the research question, articles from peer-reviewed journals, and papers issued by credible public and private organizations (e.g., government, research institutes) (lines 103-106)
-The literature selection period ranged from 2015 to 2023.(lines 106-107)
- How did the authors rate the literature? How the assessment was performed and how the authors resolved their disagreement is unclear.
We changed as follows:
-Three authors deliberated on which papers to include in the study by using a star rating system. When we were not able to reach a consensus, we further discussed the differing perspectives of each author and the selection criteria in depth.(lines 113-115)
- In the Results session, first, please provide how many papers the authors included.
We changed as follows:
-We reviewed approximately 25 papers relating to the research topic and assessed the issues using the framework as follows: (lines 169-170)
- In the Discussion session, the authors need to mention the limitations of this study.
We inserted the new sentences in the conclusion session as follows:
-In terms of the limitations of the study, we reviewed only the Korean LTCI papers concerning community care. To develop the Korean LTCI service delivery system further, it would be beneficial to study the advanced community care systems for the elderly in developed countries in the future. (lines 445-448)

This manuscript is a resubmission of an earlier submission. The following is a list of the peer review reports and author responses from that submission.
Round 1
Reviewer 1 Report
Comments and Suggestions for Authors
Thank you so much for giving me this opportunity to review this paper evaluating the long-term care insurance (LTCI) for older adults in South Korea regarding community care. The focus of this analysis is on comprehensiveness, adequacy, and integration.
I have some thoughts and comments regarding this manuscript, please find my point-by-point comments below.
Abstract
1. There should be a description of what kind of analysis the authors did for this manuscript. For example, is it a quantitative or qualitative research?
Main manuscript
1. Most of the readers may not be that familiar with South Korea's LTCI, it would be helpful to give some contexts regarding the health system/care system in South Korea.
2. On Page 3, lines 98-100, the authors mentioned the three core principles of the delivery system. It is better to have a citation for this statement. Moreover, it would be helpful for the readers to give definitions for those three principles in the context of care delivery system.
3. On page 3, line 104-106, what did the authors mean "stage 2" here, it is not very clear.
4. It is not very clear what does Figure 1 mean and how does this figure help us understand the manuscript.
5. It needs to be clarified how the authors conducted the analysis. How many researchers reviewed the framework and how did they deal with different opinions, and how did they reach a consensus?
6. What data did the authors analyze?
7. What timeframe of data did they analyze?
8. The manuscript lacks some important information regarding the research methods. This made it difficult for me to comment on the results and the conclusion derived from the study. I would recommend the authors do a thorough revision on the paper to improve the clarity of the manuscript.
Comments on the Quality of English LanguageOverall, the manuscript's English is easy to read and comprehend. However, some of the sentences are repetitive.
Reviewer 2 Report
Comments and Suggestions for Authors
Reviewer 3 Report
Comments and Suggestions for Authors
Dear authors:
The paper entitled “Analysis of Long-Term Care Insurance for Older Adults Regarding Community Care in South Korea: Focusing on Comprehensiveness, Adequacy, and Integration”. It aims to analyze the long-term care insurance (LTCI) for older adults in South Korea regarding community care focusing. It was used an analytical framework was designed for the study, focusing on comprehensiveness, adequacy, and integration. The results suggest a Korean LTCI is limited.
The study is very interesting and innovative. The outcomes are especially important to improve public policies care, headed to older people and community care namely in South Korea. The thematic falls within healthcare scope and brings an international reflection about countries policies.
I have some suggests, to contribute for improving your paper:
1. Is not clear the review methodology used: where did you make your research (i.e. data bases); who did it; when you did it? How did it?
2. In conclusion is mentioned one different aim (please uniformize it) (p.15, 499 line).
3. Reflect about the importance to maintaining figure 1 (page 5);
It really enjoy, read your paper.
I have nothing to add, and I wish you good luck towards publishing it!
Best regards.
Reviewer 4 Report
Comments and Suggestions for Authors
Dear Authors,
congratulations for the editing of your manuscript. The topic is very interesting and actual considering the current issues worldwide. However, I've only some suggestions to improve the quality of your work:
- Lines 38-40. When you talk about won, try to describe the equivalent in dollars or euros, because for an international reader could be useful and more attractive;
- Lines 561 - 2. It would be interesting to suggest detailed hypotheses of future development of subsequent studies on the same subject. Indeed, analyzing the same situation in other countries of the world, could be very interesting. I suggest you deepen your future suggestions so that you can continue on this research topic.
